# The Role of Stress Hyperglycemia on Delirium Onset

**DOI:** 10.3390/jcm14020407

**Published:** 2025-01-10

**Authors:** Ester Lagonigro, Antonella Pansini, Pasquale Mone, Germano Guerra, Klara Komici, Carlo Fantini

**Affiliations:** 1Department of Medicine and Health Sciences, University of Molise, 86100 Campobasso, Italy; elagonigro@gmail.com (E.L.); pasquale.mone@unimol.it (P.M.); germano.guerra@unimol.it (G.G.); 2ASL Avellino, 83100 Avellino, Italy; antonellapansini87@gmail.com; 3Casa di Cura “Montevergine”, 83013 Mercogliano, Italy; 4Department of Mental Health, Azienda Sanitaria Regionale Molise Antonio Cardarelli Hospital, 86100 Campobasso, Italy; carlo.fantini@asrem.molise.it

**Keywords:** diabetes, stress hyperglycemia, delirium, cognitive impairment

## Abstract

Delirium is an acute neuropsychiatric syndrome that recognizes one or more underlying causal medical conditions. Stress hyperglycemia usually refers to transient hyperglycemia associated with stress conditions such as stroke, myocardial infarction, and major surgery. Both delirium and stress hyperglycemia share common pathways, such as activation of inflammation. Stress hyperglycemia has been associated with negative outcomes, and recent studies suggested that there is an increased risk of delirium onset in patients with stress hyperglycemia. The purpose of this review is to illustrate the relationship between stress hyperglycemia and delirium. Initially, we illustrate the role of diabetes on delirium onset, summarize the criteria used for the diagnosis of stress hyperglycemia, discuss the impact of stress hyperglycemia on outcome, and focus on the evidence about the relationship between stress hyperglycemia and delirium.

## 1. Introduction

Delirium is a neurocognitive disorder characterized by the acute modification of attention awareness and cognitive function as well as behavioral abnormalities caused by an underlying medical condition [1]. Standard criteria for diagnosis have been established in the Diagnostic and Statistical Manual of Mental Disorders, 5th edition (DSM-5) and the International Classification of Disease, 10th Revision (ICD-10). The DSM-5 diagnostic criteria for delirium include acute and fluctuating disturbance of attention and awareness, disturbance in cognition, the absence of a pre-existing neurocognitive disorder and the presence of a medical condition, withdrawal, and exposure to toxins or multiple etiologies. The ICD-10 criteria include over-mentioned psychomotor deficits and sleep disturbance or sleep–wake cycle disturbance. The prevalence of delirium ranges widely from 1.4 to 70% in relationship to the screening tools applied, diagnostic criteria, and comorbidities such as dementia [2]. The clinical presentation of delirium includes three different subtypes: hyperactive, hypoactive, and mixed. These three subtypes differ in psychomotor activity. Specifically, hyperactive delirium is characterized by agitation, restlessness, hallucination, and aggression; hypoactive delirium is characterized by lethargy, inattentiveness, and motor slowness; and mixed delirium is characterized by fluctuation between the other two subtypes [2]. Delirium has been associated with an increased risk of re-hospitalizations, dependence, and institutionalizations and reduced survival [3,4].

The risk factors for delirium include predisposing or premorbid factors and precipitating factors. The former are related to the background characteristics of the patient, such as an advanced age, a high comorbidity burden, and dementia, while the latter are related to the present illness or to post-admission, such as major surgery, intensive care, or stroke [5]. Predisposing factors may also present confusing factors; for instance, an advanced age could be a risk factor for hypoactive delirium, and dementia may be an independent factor, probably because many people living with dementia could have recently been treated for delirium symptoms without a specific diagnosis of delirium [6,7,8].

Even though a variety of demographic, social, and age-related comorbidities, malnutrition, and drugs, have been identified as important risk factors for the onset of delirium [2], the underlying mechanisms remain poorly understood and have not been clarified yet. However, cerebral metabolic insufficiency, neurotransmitter imbalance, and inflammation have been suggested as potential biological mechanisms implicated in the pathophysiology of delirium [1]. Of interest, impairment of brain glucose supply induced by septic shock, impairment of glucose utilization by neurons triggered by insulin resistance, and altered brain expression of glucose transporters GLUT1 and GLUT3 may be possible pathways involved in the genesis of delirium [9]. Indeed, sepsis enhanced the transcription of several inflammatory cytokines in the brain and also altered glucose metabolism [10]. Furthermore, FDG-PET scans of hospital patients with delirium showed impaired glucose metabolism [11].

Stress hyperglycemia, known as transient hyperglycemia secondary to inflammation and neurohormonal disorders, is a common manifestation in patients with acute myocardial infarction and stroke and organ failure. Furthermore, the presence of stress hyperglycemia appears to increase the occurrence of neurological deficits, infection complications, and overall survival [12]. Given the crucial connection between brain glucose metabolism and neuronal network functioning, stress hyperglycemia may have a role in the development of delirium. Therefore, the purpose of this review is to illustrate the relationship between stress hyperglycemia and delirium. In this review, we initially discuss the role of diabetes in delirium onset, summarize the criteria used for the diagnosis of stress hyperglycemia, discuss the impact of stress hyperglycemia on outcome, and focus on the evidence about the relationship between stress hyperglycemia and delirium.

## 2. Materials and Methods

Two authors (E.L. and C.F.) performed a Medline/PubMed search regarding articles on stress hyperglycemia and delirium from inception until November 2024. A combination of the following major medical subject headings and free text terms were used: “delirium”, “stress hyperglycemia”, and “diabetes”. Additional articles were also identified by the reference list of studies included in this review. The search strategy was limited to studies published in the English language. Multiple studies reporting relevant data from the same cohort were also considered. Case reports, case series, and studies which did not provide a clear definition of delirium were not considered for this review. The quality of original research studies was graded using the Newcastle-Ottawa Scale by two reviewers (K.K. and C.F.), and disagreements were resolved by the consensus of a third reviewer (P.M.) (Appendix A).

## 3. Review Results

Overall, 544 articles were identified, and after removing duplicates, a total of 538 articles were considered for title and abstract screening. A review of titles, abstracts, and full-length articles resulted in a final selection of 59 articles for the association of diabetes with delirium, 26 articles for the definition of stress hyperglycemia and outcome, and 17 studies for the association of stress hyperglycemia and delirium (Figure 1).

## 4. Delirium and Diabetes

Chronic hyperglycemia represents a global epidemiologic burden, considering that in 2021, the global prevalence of diabetes was about 10%, and worldwide, about 537 million adults are living with diabetes [13,14]. Furthermore, an important growth in the prevalence of diabetes since 2019, accounting for about 74 million people, was revealed. Of note, about 44.7% of adults may be affected by diabetes but are underdiagnosed [15]. It has been well established that diabetes is associated with many complications, such as macrovascular and microvascular angiopathy. Macrovascular complications include stroke, coronary heart disease, and peripheral vascular disease. Among microvascular complications, neuropathy, retinopathy, and renal disease are the most frequent. All of these complication lead to poor quality of life, negative outcomes, and increased health care costs [16]. Diabetes has been associated with cognitive decline, dementia, and Parkinson’s disease [17]. Insulin resistance, amyloid beta-degradation, and impairment of protein tau functioning have been reported as possible path mechanisms involved in cognitive impairment associated with diabetes [18,19,20].

### 4.1. Delirium and Diabetes in Cardiac Surgery

In a study including 16,184 patients undergoing cardiac surgery with and without cardiopulmonary bypass, diabetes was an independent predictor of postoperative delirium and other complications, such as prolonged intensive care unit (ICU) stay, sternal instability and infection, respiratory insufficiency, reintubation, stroke, and renal dysfunction [21]. It should be mentioned that in this population, the prevalence of delirium was more common among patients undergoing coronary artery bypass grafting, with cardiopulmonary bypass accounting for about 7.9%, and beating heart surgery had a lower occurrence of postoperative delirium with a rate of 2.3% [22]. In addition, diabetes mellitus was a significant independent predictor for three postoperative outcome variables in coronary artery bypass surgery. Avoiding cardiopulmonary bypass in patients with diabetes seems to have a beneficial effect. Another data analysis reported that patients with diabetes undergoing off-pump coronary artery bypass grafting compared to those undergoing coronary artery bypass grafting with cardiopulmonary bypass surgery presented lower incidences of delirium and renal and respiratory insufficiency [23]. Diabetes was a significant independent predictor for postoperative outcome in coronary artery bypass surgery, and avoiding cardiopulmonary bypass in patients with diabetes seems to have a beneficial effect. Another study reported that diabetes was a risk factor for delirium development in patients after thoracic surgery [24]. In elderly and very elderly patients, diabetes was an independent predictor of post-cardiac surgery delirium [25,26,27,28]. Another observational prospective study concluded that the presence of cerebrovascular diseases and diabetes were predictors of delirium [29]. Diabetes was a pre-operative risk factor for the onset of postoperative delirium in patients undergoing cardiac surgery [30,31,32]. A prospective observational analysis in patients undergoing cardiac surgery reported age as an independent factor in the development of delirium but failed to identify diabetes as a significant predictor [33]. Underlying diseases such as diabetes were not directly related to postoperative delirium development in geriatric patients undergoing transcatheter aortic valve replacement and after post-orthopedic surgery [34,35]. Similar results regarding age were also reported in a study focusing on the risk of stroke and delirium after off-pump coronary artery bypass surgery [36]. A study including 329 patients who underwent open heart surgery did not find a significant role for diabetes in delirium onset [37]. However, another study found that a diagnosis of diabetes was associated with an increased risk of developing postoperative delirium (Odds Ratio (OR): 1.703; 95% CI: 1.401–2.071) [38]. Diabetes played significant role in the construction of prediction models of delirium in patients with type A aortic dissection [39].

### 4.2. Delirium and Diabetes in Orthopedic Surgery

Diabetes was associated with postoperative delirium in patients undergoing total joint replacement surgery (OR: 1.70; 95% CI: 1.15–2.47) [40]. A retrospective analysis of 388,424 primary elective total hip arthroplasty cases revealed that diabetes was associated with postoperative delirium [41]. A multicenter prospective study found that diabetes, an older age, and high neutrophil count and neutrophil lymphocyte ratio were potential markers for the prediction of delirium [42]. Comorbidities such as diabetes, sleep disorders, and frailty were independent risk factors for delirium in elderly patients undergoing hip arthroplasty [43]. A retrospective analysis using a large-scale national database including 1,228,879 total knee arthroplasty procedures identified diabetes as a risk factor for delirium [44]. The association between diabetes and delirium was confirmed in elderly patients with hip fracture [45,46,47,48,49] and in major orthopedic surgery [50]. In addition, diabetes had a significant role in the construction of predictive models for delirium onset [51,52].

### 4.3. Delirium and Diabetes in Other Clinical Settings

In patients with posterior cerebral arterial infarction, the involvement of medial occipital–temporal gyri, especially on the left side, was a pivotal factor in the onset of delirium, and diabetes was the only biochemical factor associated with delirium [53]. In patients undergoing spinal surgery, those with a central nervous system disorder (OR 6.480), previous surgical history (OR 3.499), an age older than 65 years (OR 3.390), diabetes (OR 2.981), blood transfusion (OR 2.537), and hemoglobin less than 100 g/L (OR 0.281) were significantly associated with the occurrence of postoperative delirium [54]. Furthermore, in head and neck flap surgery and lumbar spinal fusion, diabetes was a predictor of delirium [55,56,57]. Diabetes was a pre-operative risk factor in the onset of postoperative delirium in patients undergoing vascular surgery procedures [58]. The incidence of postoperative delirium in elderly patients with chronic lower limb ischemia was not significantly different among patients with diabetes and those without diabetes [59]. However, among patients with critical limb ischemia undergoing surgery, diabetes and malnutrition were important factors associated with delirium [60]. In contrast, another study associated hyper-glycemia and hypoglycemia to delirium occurrence only among non-diabetic patients [61]. Furthermore, diabetes was not associated with postoperative delirium in patients undergoing microvascular decompression [62]. Diabetes was not identified as a predictor for delirium during an emergency admission to hospital care, analyzing data from routine primary care records [63]. However, incorrect electronic records related to diabetes encoding did not influence the prediction model for delirium development [64]. A machine learning approach also identified diabetes as a significant factor for the detection of delirium [65].

Among 560 patients aged over 65 years admitted at internal medicine wards, cognitive impairment on admission, diabetes, chronic kidney failure, and the male gender were significantly associated with the development of delirium during hospitalization [66]. Uncontrolled diabetes was identified as a risk factor for delirium in patients admitted to a cardiac intensive care unit. Consequently, patients with delirium presented longer intensive care unit stay and a higher risk for mortality [67]. An older age, diabetes, American Society of Anesthesiologists (ASA) scores, and COPD were predictors of delirium in adults admitted at the intensive care unit [68]. About a quarter of critically ill surgery patients experience delirium and diabetes together with other characteristics, such as benzodiazepine use, mechanical ventilation, and severity of disease score had a significant role on the onset of delirium [69]. Diabetes was significant in prediction models of delirium in patients admitted at the intensive care unit [70,71,72].

Diabetes was also reported as a risk factor for delirium in patients after liver section surgery [73], gastrointestinal cancers [74,75], or reconstructive surgery [76]. Pre-stroke apathy and diabetes were predictors of delirium after stroke [77]. Of interest, a higher cardiovascular risk score was associated with postoperative delirium partially mediated by tau protein [78]. A recent study reported that diabetes is an independent risk factor for delirium superimposed on dementia in elderly patients in a comprehensive ward [79].

Appendix A summarizes the above-mentioned studies.

## 5. Stress Hyperglycemia Diagnosis and Outcome

An increased level of blood glucose is not only chronic, but it could also be an acute physiological response to a physiological or pathological stress. Stress hyperglycemia usually refers to transient hyperglycemia during an acute illness, and it can be observed in patients with a previous diagnosis of diabetes, in patients with newly diagnosed diabetes, as well as in non-diabetic patients [9,80,81]. In the present literature, the description of stress hyperglycemia is derived by individual studies in a heterogenous way, and most commonly, stress hyperglycemia refers to blood glucose drawn on admission with a glycemia range of 108–180 mg/dL or fasting blood glucose drawn the next morning on admission with a range of 110–141 mg/dL [82]. Although there is no clear definition for stress hyperglycemia, it has been proposed to diagnose stress hyperglycemia if (a) the hospital-related fasting glucose is >6.9 mmol/L or random glucose levels are >11.1 mmol/L without evidence of previous diabetes or if (b) there is pre-existing diabetes with the deterioration of glycemic control [9].

In addition, recent studies have introduced a stress hyperglycemia ratio (SHR) that combines admission glucose and the estimated average glucose derived from glycosylated hemoglobin (HbA1c). Considering that HbA1c reflects the mean glucose status over the preceding three months, it has been suggested that the SHR is able to express stress hyperglycemia more accurately.

Stress hyperglycemia has been associated with an increased risk of diabetes in the ICU (OR 3.48; 95% CI 2.02–5.98) [83]. Furthermore, in the ICU setting, glycemic control has been described as an important factor associated with poor outcomes [84], and stress hyperglycemia was characterized by a 2-fold increased risk for mortality [85]. Stress hyperglycemia has been associated with a longer ICU length of stay in patients recovered after endoscopic intracerebral hemorrhage evacuation [86]. Critically ill patients with cardiogenic shock with stress hyperglycemia required more frequent mechanical ventilation and were characterized by higher rates of mortality [87]. Hyperglycemia in critically ill patients with COVID-19 was associated with longer ICU hospitalization [88]. Stress hyperglycemia was associated with increased three-month mortality in ICU patients with pulmonary hypertension [89]. A retrospective study including 3887 ICU patients found that the SHR was linked to ICU mortality and 1-year all-cause mortality. This association was more robust among non-diabetic patients rather than patients with diabetes [90]. Of interest, another retrospective study of 5564 patients admitted at a cardiac ICU reported a U-shaped association between the SHR and short-term mortality [91]. An SHR value of 0.95 represents the inflection point for a negative outcome. Furthermore, 2312 critically ill patients with sepsis were included in a retrospective cohort study, which showed an increased 28-day all-cause mortality in patients with a higher SHR [80]. In 451 patients with moderate–severe COVID-19 infection, the SHR was a better predictor of outcome compared to admission blood glucose irrespective of the pre-existing chronic glycemic status [92]. In patients with sepsis, a J-shaped association between stress hyperglycemia and mortality has been suggested [93].

In both American and Chinese cohorts, patients with an elevated SHR are characterized by increased 1-year and long-term all-cause mortality [94]. This association was more pronounced among non-diabetic patients compared to patients with diabetes. In contrast, data from another study indicate that, especially among patients with diabetes with myocardial infarction and non-obstructive coronary disease, the SHR was a better predictor of poor prognosis during a mean follow-up of about 3.5 years [95]. A recent systematic review and meta-analysis study which investigated the prognostic impact of the SHR in patients with acute myocardial infarction reported that a higher SHR was characterized by major adverse cardiovascular and cerebrovascular events (HR = 1.7; 95% CI = 1.42, 2.03), higher long-term mortality (HR = 1.64; 95% CI = (1.49–1.8)), and higher in-hospital all-cause mortality (OR = 3.87; 95% CI = 2.98–5.03) [96]. Of note, in patients with acute myocardial infarction, the use of sodium glucose co-transporter-2 inhibitors (SGLT2-I) was a significant predictor of reduced inflammatory response independently of admission glycemia and other confounders such as age and renal function [97]. Data from 1099 patients with ST-elevation myocardial infarction (STEMI) who have undergone percutaneous coronary intervention (PCI) report that stress hyperglycemia should be considered as an important prognostic factor regardless of the presence or absence of diabetes [98].

The incidence of stress hyperglycemia ranged between 21 and 27% in patients with ischemic stroke [99]. The SHR calculated from the fasting glucose and fasting glucose (mmol/L)/HbA1c% was the best predictor for poor outcome at three months in patients affected by acute ischemic stroke undergoing intravenous thrombolysis [100], and stress hyperglycemia was related to a higher risk of 90-day stroke recurrence [101]. In non-diabetic patients, the SHR was related to an increased risk of stroke recurrence and all-cause mortality [102]. Of interest, a significant correlation between stress hyperglycemia and vascular cognitive impairment in patients with ischemic stroke has been suggested [103].

Appendix A summarizes the above-mentioned studies.

### Stress Hyperglycemia and Delirium

In patients undergoing general thoracic surgery, abnormal glycemia levels above 16.5 mmol/L has been linked to the development of postoperative psychiatric disorders. Even though in this study, postoperative psychiatric disorders included psychosis, minor depression, adjustment disorders, and panic attack, delirium was the most common disorder as it was present in about 44% of patients [104]. A retrospective study reported that in the intensive care unit, patients with hyperactive delirium presented higher glucose levels measured by blood gas analysis compared to patients without delirium [105]. Hyperglycemia defined as a blood glucose level above 8 mmol/l was a significant determinant of delirium in non-diabetic patients admitted to the intensive care unit (ICU) [61].

Another study which investigated the outcome of metabolic disturbance on delirium in patients undergoing coronary artery bypass grafting surgery reported that patients with delirium experienced a hyperglycemic state at the intraoperative (*p* = 0.004), intubation (*p* = 0.03), and extubation periods (*p* = 0.02) [106]. Intraoperative hyperglycemia, defined as blood glucose levels higher than 150 mg/dl in patients undergoing elective surgery, was independently associated with postoperative delirium (OR 3.86 CI 95% 1.13, 39.49), and a relative duration of hyperglycemia also played a significant role in the development of postoperative delirium among non-diabetic patients [107].

Blood glucose variability measured by monitoring the blood glucose level every 2–6 h in 48 h intervals and assessed by the standard deviation of all measurements was a significant discriminator of delirium incidence among patients with acute aortic dissection (AUC = 0.763; 95% CI = 0.704–0.821) [108]. A predictive scoring system applied in the cardiac surgery setting, which also included diabetes treatment and pre-operative fasting glucose among other variables showed excellent accuracy with an area under the receiver operator curve of 0.83, a specificity of 92%, and sensitivity of 60% [109]. Increased glucose levels were described in patients that developed delirium in the rehabilitation setting [110]. A blood glucose level ≥ 8.05 mmol/L was a significant factor for delirium prediction in patients with hip fractures [111]. In patients undergoing hip fracture surgery, pre-operative pre-injury stair-climbing capacity and pre-operative fasting blood glucose may identify patients at a high risk for the development of delirium [112]. Glycemic variability in patients admitted at non-intensive care unit wards was correlated to an increased length of hospitalization, poor survival, and delirium [113]. Of interest, in 1951 patients with diabetes undergoing coronary artery bypass grafting, higher levels of mean blood glucose (OR = 3.703; 95% CI: 1.743–7.870), a mean absolute glucose >0.77 mmol/L/h (OR = 1.754; 95% CI: 1.235–2.490), and a glycemic lability index >2.6 (mmol/L)^2^/h (OR = 1.458; 95% CI: 1.033–2.058) were associated with higher odds of postoperative delirium [114]. Elevated levels of pre-operative fasting plasma glucose were associated with delirium in non-diabetic patients undergoing total hip replacement, and this association was in part mediated by tau protein [115]. Despite the fact that isolated impaired fasting glucose did not reflect any clinical manifestation in patients undergoing cardiopulmonary bypass surgery, higher levels of neuron-specific enolase were detected, suggesting a connection between stress hyperglycemia and brain metabolism [116].

In patients with diabetes, hypoglycemia was associated with delirium (adjusted OR: 2.78; 95% CI: 1.71–6.32) [61,117]. Data from a randomized double-blind controlled trail revealed that patients randomized to tight glucose control were more likely to be diagnosed with delirium than those assigned to routine glucose control (relative risk: 1.89; 95% CI, 1.06–3.37) [118]. Relative hypoglycemia, defined as at least a 30% reduction in baseline glycemia, was significantly associated with the occurrence of delirium among critically ill patients [119]. Higher glucose levels were present in critically ill patients who developed delirium [105], and intraoperative hyperglycemia was significantly associated with delirium [120].

Furthermore, another study that included 23,532 patients with diabetes who underwent non-cardiac surgery and analyzed both chronic hyperglycemia (HbA1c > 6.5% within three months before surgery) and acute hyperglycemia (fasting blood glucose > 140 mg/dL or random glucose > 180 mg/dL within 24 h before surgery) found a significant correlation only between acute hyperglycemia and delirium [121]. Another study that included a large number of participants, specifically 203,787 patients who underwent non-cardiac surgery, revealed that hyperglycemia is consistently associated with delirium regardless of the presence of diabetes [122].

Few studies considered stress hyperglycemia, expressed as the SHR and delirium. An example is a cohort study that observed an increased incidence of delirium both in older hospitalized patients with lower SHRs and higher SHRs. Actually, the incidence of delirium was higher in the first and in the third tertiles of SHR values and only among patients with HbA1c < 6.5% [123].

Appendix A summarizes the included studies.

## 6. Discussion

Diabetes has been associated with the onset of postoperative delirium in patients undergoing cardiac surgery, orthopedic surgery, gastro-intestinal surgery, and other clinical settings, such as the intensive care unit and stroke and geriatric wards. The description of stress hyperglycemia is heterogeneous; however, a clinical condition of transient hyperglycemia during an acute illness has been referred to as stress hyperglycemia [9]. Stress hyperglycemia has been described in up to 27% of patients [99], and it has been associated with an increased risk of diabetes, major adverse cardiovascular events, in-hospital mortality, and poor long-term survival [83,94,96]. Acute hyperglycemia, glucose variability, and SHR have been associated with delirium in both non-surgery and surgery clinical settings, and this association has been described regardless of the presence or absence of diabetes [73,94,114,115,122,123].

A previous systematic review study concluded that perioperative hyperglycemia increased the risk of postoperative delirium in patients undergoing surgery [124]. It should be mentioned that stress hyperglycemia usually occurs in the first few days after an acute illness, and the increase in glycemia levels may initially have a protective role. Indeed, stress hyperglycemia has been suggested as a survival response in the central nervous system, and mild-to-moderate stress hyperglycemia may protect against apoptosis by favoring angiogenesis and increasing neuronal plasticity [125]. On the other hand, an enhanced activation of the inflammatory and neuro-hormonal axis, which occurs during the acute illness, may induce higher stress hyperglycemia which, in turn, may activate the mechanisms of oxidative stress and cerebrovascular endothelial dysfunction and lead to delirium development [125]. Furthermore, chronic hyperglycemia may result in cellular adaptation against glucose fluctuation and the down-regulation of glucose transporters. Of interest, a recent study of cardiac ischemia–reperfusion animal models of delirium reported that pre-treatment with liraglutide, a long-acting glucagon-like peptide-1 (GLP-1), was protective against delirium behaviors via microglia mitophagy, neuroinflammation, and preserving synaptic integrity [126]. Furthermore, metformin use was associated with a lower risk of delirium onset (Hazard Ratio: 0.77–0.88) [127].

Future experimental studies should clarify the potential mechanisms that link stress hyperglycemia with delirium development. In addition, the impact of stress hyperglycemia on delirium development should be expanded in larger studies in different clinical settings to test the clinical utility of stress hyperglycemia on delirium development.

### 6.1. Delirium and Inflammation

The pathogenesis of delirium remains unclarified; however, different pathways have been proposed, such as the reduction in cerebrovascular blood flow, the modification of cerebral metabolism, dopamine excess, and the acetylcholine deficiency alternation of the sleep–wake cycle and inflammation. The activation of systemic inflammatory cascade and the dysregulation of cytokines may be crucial for the onset of delirium [128]. Systemic inflammation triggers the release of tumor necrosis factors (TNFs) IL-1B and IL-6 by the activation of leucocytes, macrophages, and monocytes. In addition, the activation of Toll-like receptors on macrophage cells residing in the choroid plexus and circumventricular organs and the activation of IL-1 receptors of the endothelial cells of brain venules enhance the production of pro-inflammatory cytokines, which may compromise the blood–brain barrier [1,129]. IL-1 acts directly on multiple brain cells, such as astrocytes, and facilitates the homing and recruitment of leucocytes to the brain, which contributes to cognitive disfunction. Indeed, increased levels of cerebrospinal fluid and IL-1B and IL-8 were reported in patients with hip fracture and delirium compared to patients who did not have delirium [130,131]. Furthermore, in response to acute systemic inflammation, microglial cells enhance the synthesis of IL-1, and their activation has been associated with delirium and cognitive decline [132,133].

Experimental models revealed that the administration of interleukins may induce symptoms similar to delirium [134]. In elderly hospitalized patients, delirium onset was significantly associated with cytokines IL-6 and Il-8, even after adjusting for confounders such as age, infection, and baseline cognitive impairment [135]. Of note, Il-8 levels are higher in the days before delirium onset, whereas IL-6 has been associated with hyperactive delirium [136]. In patients aged 75 years and over admitted at an acute ward, a low baseline IGF-1 level was a risk factor for incident delirium [137]. Furthermore, differences in serum IL-8, IL-10, and CRP concentrations were associated with postoperative delirium [138]. A meta-analysis study confirmed that IL-6 was a consistent predictor for delirium in the surgical setting [139]. High serum cortisol levels, age, diabetes, and a prolonged time of surgery were associated with postoperative delirium in patients undergoing coronary artery bypass graft surgery [140]. High levels of c-reactive protein improved the prediction of post-stroke delirium when also considering diabetes [141]. An increased systemic immune inflammation index was a valuable biomarker for the prediction of delirium [142] (Figure 2 summarizes the relationship between inflammation and delirium).

### 6.2. Stress Hyperglycemia and Inflammation

Inflammation is a wide group of physiological responses to a foreign organism, including human pathogens, dust particles, and viruses. Inflammation involves a long chain of molecular reactions and cellular activity [143]. Stress hyperglycemia is characterized by transient hyperglycemia during acute and/or severe illness, related to an adaptive mechanism to provide energy for the organism during physiological or pathological stress, although this hyperglycemic state may be associated with unfavorable clinical outcomes, particularly in patients facing prolonged stress or in patients affected by multiple comorbidities [9]. The complex interplay between insulin resistance, beta-pancreatic cells’ secretory defects, growth hormone, cortisol, cytokines, and catecholamines may affect the development of stress hyperglycemia. Stress hyperglycemia and inflammation have some common pathogenic mechanisms since both are related with cytokines’ release, such as tumor necrosis factor-α (TNF-α), which might promote gluconeogenesis by stimulating glucagon production, as well as inhibit post-receptor insulin signaling [144]. Hyperglycemia and inflammation could be involved in a vicious cycle in which hyperglycemia leads to further hyperglycemia, since hyperglycemia induces a higher release of cytokine, a higher inflammatory response, and a higher oxidative stress response [145]. Conversely, when this vicious cycle is interrupted, both inflammation and hyperglycemia normalize, so the normalization of glycaemia is associated with the resolution of the inflammatory response [9].

Indeed, an abnormal cortisol concentration has been associated with delirium, suggesting an impairment of hypothalamic–pituitary–adrenal function [140]. Furthermore, alterations in cortisol levels are present in patients with pneumonia, urinary tract infections, sepsis, anxiety, and cognitive impairment [146,147,148]. Of interest, patients with mild cognitive impairment who developed postoperative delirium presented higher plasma levels of IL-2 and cortisol [149]. Other studies have reported that patients with mild cognitive impairment presented a higher likelihood of developing delirium, and patients with apparently normal cognitive function before surgery who experienced postoperative delirium tended to be more likely to be diagnosed with mild cognitive impairment [150,151,152,153].

Moreover, insulin resistance promotes some catabolic processes, for instance, lipolysis, which exacerbates the inflammatory state. In this context, lipolysis and lipotoxicity, as well as glucotoxicity and inflammation, could be considered key components of insulin resistance associated with acute illness [154].

Different biomarkers were used to confirm the clinical suspicion of inflammation, such as the white blood cell count (WBC), neutrophil-to-lymphocyte ratio (NLR), C-reactive protein (CRP), pro-calcitonin (PCT), erythrocyte sedimentation rate (ESR), and D-dimer. These elements were considered in a multicenter and retrospective study involving 1631 patients that analyzed the relationship between SHR and inflammation in patients with diabetes with pneumonia on admission, showing a significant increase in all inflammatory indexes in patients with higher SHRs (4th quartile, which means SHR > 1.092) [155]. In addition, PCT, NLR, and SHR were considered in a recent cohort-retrospective study including 406 patients with SCAP (Severe Community-Acquired Pneumonia). All of these elements were associated with mortality (*p*-value < 0.001) [156]. NLR acted as a mediator in the relationship between SHR and survival and vice versa. In other words, a more robust inflammatory response might exacerbate hyperglycemia and vice versa, potentially worsening the prognosis. Thus, inflammation and stress response (including stress hyperglycemia) should not be regarded as independent phenomena, but rather, they should be reputed as two phenomena that are involved in a complex relationship (Figure 3 summarizes the relationship between stress hyperglycemia, inflammation, and delirium).

## 7. Conclusions

Stress hyperglycemia usually refers to transient hyperglycemia associated with an acute illness. Despite the fact that the diagnosis of stress hyperglycemia is heterogenous and specific criteria are still lacking, hyperglycemia is mostly considered if, during hospitalization, the fasting glucose is >6.9 mmol/L, the random glucose is >11.1 mmol/L, or there is a deterioration in glycemic control in patients with pre-existing diabetes.

Diabetes and stress hyperglycemia are associated with an increased risk of cardiovascular events, prolonged hospitalization, and poor survival and an increased risk of delirium in the surgery setting, the intensive care unit, and in elderly hospitalized patients. Stress hyperglycemia and delirium share common mechanisms, and the activation of inflammatory cascade may be a cross-linked pathway. Future experimental and clinical studies are necessary to better explore the role of stress hyperglycemia in delirium onset and to investigate potential mechanisms and management options.

## Figures and Tables

**Figure 1 jcm-14-00407-f001:**
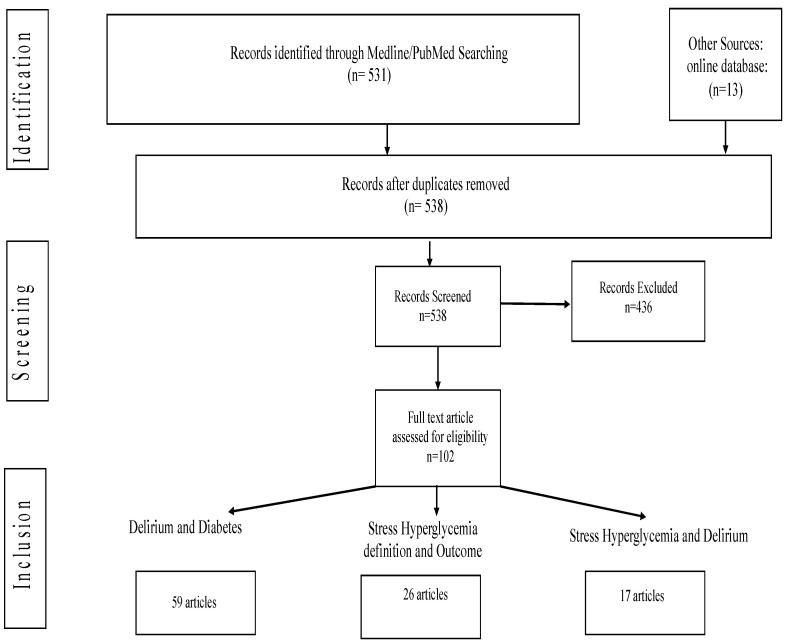
A flowchart of articles included in the review.

**Figure 2 jcm-14-00407-f002:**
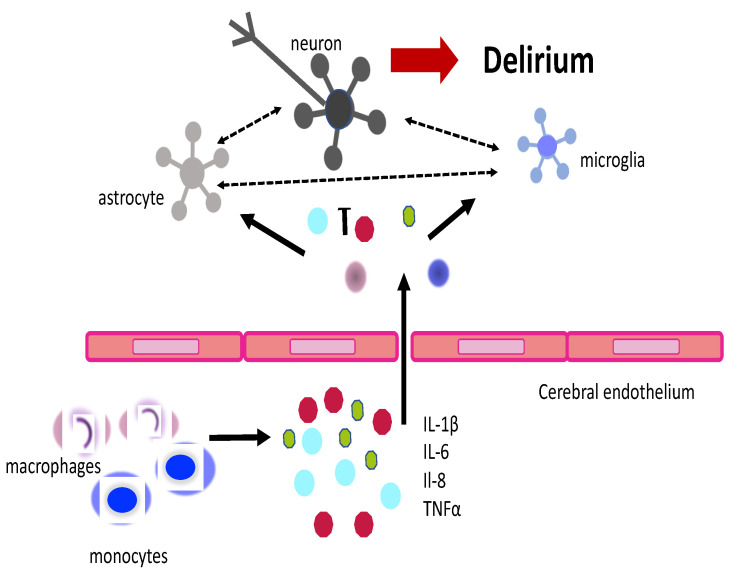
Relationship between delirium and inflammation. Color dots represent IL-1B, IL-6, IL-8, TNFα; arrows indicate the sequences of events.

**Figure 3 jcm-14-00407-f003:**
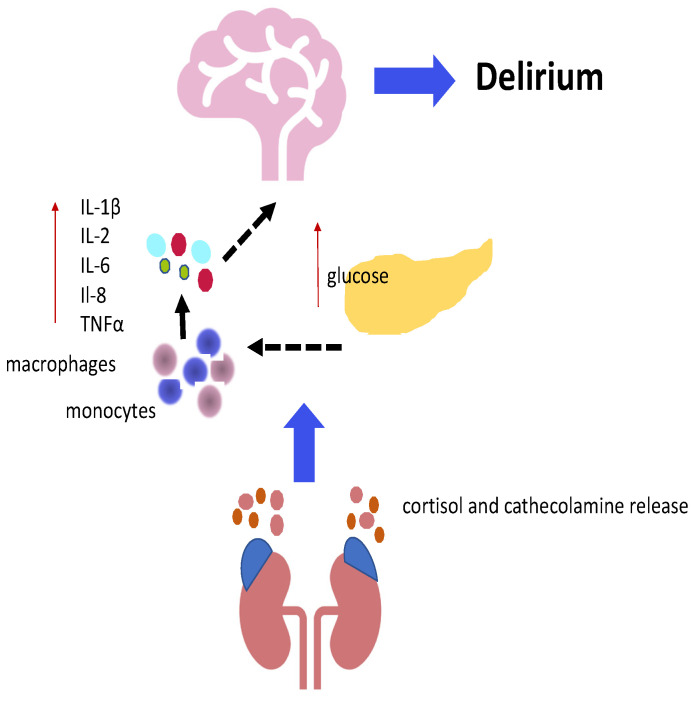
The relationship between stress hyperglycemia, inflammation, and delirium. Color dots represent IL-1B, IL-6, IL-8,TNFα. Blue and black arrows indicate the sequences of events, red arrows indicate increased production.

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
