# Peer review of "The Role of Stress Hyperglycemia on Delirium Onset"

_jcm, 2025, doi:10.3390/jcm14020407_

Round 1
Reviewer 1 Report
Comments and Suggestions for Authors
The paper deals with an interesting and inmportant topic. This deserves further attention.
There is no real research question. This should be at the end of the section 'introduction'.
There is no search strategy (MeSH terms, inclusion and exclusion criteria for manuscripts). This increases the risk for bias. A flowchart would be helpful.
The PRISMA guideline list should be followed, even if a meta-analysis is not attempted. This allows increases the readability of the article.
This would be the main part of the section 'methods'.
The quality of the included papers should be evaluated. A possible tool is the Newcastle-Ottawa-Scale. This is of major importance since the diagnostic criteria and tools for delirium might differ between authors.
The role of preoperative mild cognitive impairment in surgical cases (which is not always included in the work-up of non-acute patients) as well is in acute inflammatory disorders ( which could be evaluated in survivors, after recovery) is not taken into account.
The results should be quantified and tabulated as much as possible. This would give readers an oversight of the results. This would be the main part of the section 'results'
Cardiac surgery with use of an extracorporeal circulation deserves a separate section because of the possible role of this device.
The discussion and the conclusion should be adapted to the findings.
Comments on the Quality of English Language
There are several typo's that need correction. There are no other major language issues.
Author Response
Reviewer #1: The paper deals with an interesting and inmportant topic. This deserves further attention.
There is no real research question. This should be at the end of the section 'introduction'.
Reply:Thank You for the positive evaluation of our work. In the revised version of our paper we clarified the research question and the aim of our review. At the end of the section introduction we added the following text: ''Given the crucial connection between brain glucose metabolism and neuronal network functioning, stress hyperglycemia may have a role in the development of delirium. Therefore, the purpose of this review is to illustrate the relationship between stress hy-perglycemia and delirium. In this review we initially discuss the role of diabetes on delirium onset, summarize the criteria used for the diagnosis of stress hyperglycemia, discuss the impact of stress hyperglycemia on outcome and focus on the evidence about the relationship between stress hyperglycemia and delirium''.
There is no search strategy (MeSH terms, inclusion and exclusion criteria for manuscripts). This increases the risk for bias. A flowchart would be helpful.
The PRISMA guideline list should be followed, even if a meta-analysis is not attempted. This allows increases the readability of the article.
This would be the main part of the section 'methods'.
The quality of the included papers should be evaluated. A possible tool is the Newcastle-Ottawa-Scale. This is of major importance since the diagnostic criteria and tools for delirium might differ between authors.
Reply: Our article is a narrative review however, following the Reviewer's suggestion we added ''materials and methods section'', and ''results section''. MeSH terms, key words, free terms used inclusion and exclusion criteria were added. A flowchart regarding article selection was added in the text and also original research articles were graded with Newcastle-Ottawa-Scale. Please check in the revised version and supplementary Table 1.
The role of preoperative mild cognitive impairment in surgical cases (which is not always included in the work-up of non-acute patients) as well is in acute inflammatory disorders ( which could be evaluated in survivors, after recovery) is not taken into account.
Reply: Thank You for the interesting suggestion. In the discussion section we discussed this important point and added the following paragraph with appropriate references: ''Of interest in patients with mild cognitive impairment who developed post-operative delirium presented higher plasma levels of IL-2 and cortisol [149]. Other studies have reported that patients with mild cognitive impairment presented higher like-hood of developing delirium and patients with apparently normal cognitive function before surgery, who experienced post-operative delirium tended to be more likely diagnosed with mild cognitive impairment [150-153]''.
The results should be quantified and tabulated as much as possible. This would give readers an oversight of the results. This would be the main part of the section 'results'
Cardiac surgery with use of an extracorporeal circulation deserves a separate section because of the possible role of this device.
The discussion and the conclusion should be adapted to the findings.
Reply:The results were divided in the following paragraphs: delirium and diabetes in cardiac surgery, delirium and diabetes in orthopedic surgery, delirium and diabetes in other clinical settings, stress hyperglycemia definition and outcome, stress hyperglycemia and delirium. In the paragraph: delirium and diabetes in cardiac surgery are provided details of surgery procedures with beneficial effects. The discussion and conclusion section were added and adapted to the results. Please check the text.
We wish to Thank the Reviewer for his/her constructive criticism and suggestions which helped us to ameliorate our work.

Reviewer 2 Report
Comments and Suggestions for Authors
The paper is well-prepared. The authors have presented the connection between delirium and stress hyperglycemia. They have also drawn attention to the role of inflammation in this process.
However, the manuscript requires the addition of at least 2 figures to show the mechanisms described. The first figure should concern the inflammatory process in delirium. I also think that the chapter "1.1. Delirium and Inflammation" should be described in more detail, including the role of specific cells involved in the inflammation. The second figure should show how stress hyperglycemia can affect the development of inflammation and ultimately the development of delirium.
Author Response
Reviewer # 2
The paper is well-prepared. The authors have presented the connection between delirium and stress hyperglycemia. They have also drawn attention to the role of inflammation in this process.
However, the manuscript requires the addition of at least 2 figures to show the mechanisms described. The first figure should concern the inflammatory process in delirium. I also think that the chapter "1.1. Delirium and Inflammation" should be described in more detail, including the role of specific cells involved in the inflammation. The second figure should show how stress hyperglycemia can affect the development of inflammation and ultimately the development of delirium.
Reply: Thank You for the positive comments regarding our work. Following the suggestion of the Reviewer, we added 2 figures which summarize the role of inflammation on delirium onset, and the connection between stress hyperglycemia, inflammation and delirium. In addition, the delirium and inflammation section was extended and information regarding cells such as, macrophages, monocytes, microglia, astrocytes were provided. Please check the revised version of our manuscript, page 9 and 11.
We wish to Thank the Reviewer for his/her constructive criticism and suggestions which helped us to ameliorate our work.

Round 2
Reviewer 1 Report
Comments and Suggestions for Authors
The information offered in the paper is important and clinically relevant. The figures clarify the involved mechanisms.
However, to improve the readability of the paper, the main results of the included studies shoulf be tabulated for isk factors, outcome and the associated odds (or isk) ratios, 95% confidence intervals and p-values.
Some tpo's should be corrected
Author Response
Reviewer# 1
The information offered in the paper is important and clinically relevant. The figures clarify the involved mechanisms.
However, to improve the readability of the paper, the main results of the included studies shoulf be tabulated for isk factors, outcome and the associated odds (or isk) ratios, 95% confidence intervals and p-values.
Some tpo's should be corrected
Reply: Thank You for the suggestions. In the revised version of our manuscript we provided three tables where the results of the studies included in our review were tabulated regarding: type of study, study design, aim, population, definition of stress hyperglycemia, diagnosis of delirium, main results, other results and risk factors. Based on data availability the relationship between diabetes and delirium, stress hyperglycemia and delirium was expressed also in OR, HR, 95% CI and p-value. Please check supp. tables 2-4. Typos were corrected.
Thank You for the comments. We hope that we addressed all the comments and suggestions.
Reviewer 2 Report
Comments and Suggestions for Authors
The authors significantly corrected the manuscript according to the reviewer's suggestions. Recently, I recommend the article for publication.
Author Response
Reply: Thank You for the positive comments regarding our work